# Prenyl Transferases Regulate Secretory Protein Sorting and Parasite Morphology in *Toxoplasma gondii*

**DOI:** 10.3390/ijms24087172

**Published:** 2023-04-12

**Authors:** Qiang-Qiang Wang, Kai He, Muhammad-Tahir Aleem, Shaojun Long

**Affiliations:** 1National Key Laboratory of Veterinary Public Health Security, School of Veterinary Medicine, China Agricultural University, Beijing 100193, China; B20193050436@cau.edu.cn (Q.-Q.W.); m15537931996@163.com (K.H.); 2Center for Gene Regulation in Health and Disease, Department of Biological, Geological and Environmental Sciences, College of Sciences and Health Professions, Cleveland State University, Cleveland, OH 44115, USA; m.aleem@csuohio.edu

**Keywords:** *Toxoplasma gondii*, conditional knockdown, TgFT, TgGGT-1, TgGGT-2

## Abstract

Protein prenylation is an important protein modification that is responsible for diverse physiological activities in eukaryotic cells. This modification is generally catalyzed by three types of prenyl transferases, which include farnesyl transferase (FT), geranylgeranyl transferase (GGT-1) and Rab geranylgeranyl transferase (GGT-2). Studies in malaria parasites showed that these parasites contain prenylated proteins, which are proposed to play multiple functions in parasites. However, the prenyl transferases have not been functionally characterized in parasites of subphylum Apicomplexa. Here, we functionally dissected functions of three of the prenyl transferases in the Apicomplexa model organism *Toxoplasma gondii* (*T. gondii*) using a plant auxin-inducible degron system. The homologous genes of the beta subunit of FT, GGT-1 and GGT-2 were endogenously tagged with AID at the C-terminus in the TIR1 parental line using a CRISPR-Cas9 approach. Upon depletion of these prenyl transferases, GGT-1 and GGT-2 had a strong defect on parasite replication. Fluorescent assay using diverse protein markers showed that the protein markers ROP5 and GRA7 were diffused in the parasites depleted with GGT-1 and GGT-2, while the mitochondrion was strongly affected in parasites depleted with GGT-1. Importantly, depletion of GGT-2 caused the stronger defect to the sorting of rhoptry protein and the parasite morphology. Furthermore, parasite motility was observed to be affected in parasites depleted with GGT-2. Taken together, this study functionally characterized the prenyl transferases, which contributed to an overall understanding of protein prenylation in *T. gondii* and potentially in other related parasites.

## 1. Introduction

*Toxoplasma gondii*, as an obligate intracellular parasite and a model of apicomplexan parasites, can infect almost all nucleated cells of warm-blooded animals [1]. Although the parasite generally causes mild symptoms in immunocompetent people, infections can lead to severe toxoplasmic encephalitis in lymphoma and AIDS patients [2]. Additionally, *T. gondii* can cause severe neurological diseases, blindness and even death from congenital infections [3,4]. Currently, several compounds (inhibitors) showed good bioactivity against *T. gondii* in a mouse model during the acute and chronic infection, such as bumped kinase inhibitors (BKI 1294) [5], electron transport chain inhibitors (ELQ271) [6], fatty acid synthesis inhibitors (Triclosan) [7], folic acid synthesis inhibitors (JPC-2056) [8], histone modification inhibitors (FR235222) [9], protein synthesis inhibitors (Clindamycin) [10] and farnesyl diphosphate/geranyl geranyl-diphosphate synthase enzyme inhibitors (Atorvastatin) [11]. However, these chemotherapeutic choices for curing toxoplasmosis are not clinically applicable. It is highly important for the community to identify potential drug targets, such as enzymes that are able to control diverse biological processes.

Plenty of proteins require lipid modification by attaching an isoprenoid lipid to execute its normal functions, such as cell attachment, differentiation and growth [12]. Completion of the kind of lipid modification process (prenylation) requires enzymes of three types. The well-known enzymes include three types: protein farnesyl transferase (FT), geranylgeranyl transferase type-I (GGT-I) and geranylgeranyl transferase type-II (GGT-II/RabGGTase) [13,14]. Under the catalyzation of FT and GGT-I, the 15-carbon isoprenoid and 20-carbon isoprenoid can be transferred from their substrate farnesyl diphosphate (FPP) or geranylgeranyl diphosphate (GGPP) to the target protein or short peptide, respectively, the C−terminal of which has one recognition motif (CaaX). In this motif box, C symbolizes the cysteine, and aa means two typically aliphatic residues. As for the X of CaaX, it is different for the two enzymes, where X is usually glutamine, threonine, cysteine, serine or methionine recognized by FTase, and X is leucine specificity for GGT-I [15,16,17,18,19,20,21,22]. In contrast to FT and GGT-I, GGT-II is more distinctive and uniquely modifies the family of Rab proteins [23]. GGT-II can transfer 20-carbon isoprenoid to both cysteine residues of CxC- or CC-comprising Rab proteins [24]. Furthermore, synthetic short peptides cannot be recognized by GGT-II [25].

Recently, protein farnesyl transferase and geranylgeranyl transferase type-I exhibited good potency and had been regarded as promising drug targets for preventing malaria [26,27,28,29]. However, knowledge about protein prenyl transferases in *T. gondii* is very limited. In this research, we clarified the effects of the three protein prenyl transferases of *T. gondii* on various organelles. Moreover, the replication and gliding ability of parasites were detected after the degradation of TgFT, TgGGT-1 and TgGGT-2. The current study first uncovers details of prenyl transferases functions on *T. gondii* and deepens the understanding of prenyl transferases, which prompts the identification of new drug targets in *T. gondii*.

## 2. Results

### 2.1. Conservation and Phylogenetic Analysis

From the web server (https://toxodb.org/toxo/app) (accessed on 2 June 2022), FT, GGT-1 and GGT-2 were identified as the only three prenyl transferases in *T. gondii*. Among the three prenyl transferases, FT (TGGT1_200370) was predicted as not essential (−0.53) for the parasite. GGT-1 (TGGT1_278230), and GGT-2 (TGGT1_316270) were predicted as essential (−4.25, −4.27, respectively) for the parasite (Figure 1a). For the phylogenetic analyses targeted to FT, GGT-1 and GGT-2 in the genome of *T.gondii*, we found that there were five functional domain regions with inconsistent lengths (Figure 1b). Prenyl transferase orthologs were present in all available apicomplexan genomes, and their alignment recapitulated the known relationship among these species (Figure 1c). Furthermore, according to the conversation heatmap across the Alveolate, the existence or absence of orthologues for the three proteins found in 121 alveolate taxa was displayed in Figure 1d, with proteins clustered according to their phylogenetic distributions. Similar homologous results were also revealed in other species, including ciliophora, dinoflagellates and perkinsozoa. Furthermore, the complete and available information of each protein homologue can be found in Appendix A.

### 2.2. Localization of FT, GGT-1 and GGT-2

With the purpose of examining the localization of FT, GGT-1 and GGT-2 in *T. gondii*, the C-terminals of FT, GGT-1 and GGT-2 added the HA tag through endogenously tagging using the CRISPR/Cas9 system. The model of the C-terminal of gene tagging could be clearly observed in Figure 2a. The IFA was carried out to confirm if they were tagged with HA successfully. The results showed that FT uniformly dispersed in the whole cytoplasm of parasites. GGT-1 and GGT-2 localized in the entire cytoplasm with some major distribution in the basal part of parasites (Figure 2b). 

### 2.3. Construction of FT-AID, GGT-1-AID and GGT-2-AID Strains

The Auxin-inducible Degron (AID) system was used in this study to construct conditional knockdown parasites for investigating the functions of FT, GGT-1 and GGT-2 in *T. gondii*. The proteins of *T. gondii* could be effectively degraded after adding the drug IAA, and the model of protein degradation was shown in Figure 3a. Here, the C-terminals of FT, GGT-1 and GGT-2 were successfully tagged with AID-ty fragments, and we could observe that FT, GGT-1 and GGT-2 were completely degraded when the parasites were treated with IAA (500 μM) for 24 h (Figure 3b).

### 2.4. Effects of FT, GGT-1 and GGT-2 on Proliferation of T. gondii

The FT-AID, GGT-1-AID and GGT-2-AID were treated with IAA for 24 h to observe their effects on proliferation of parasites, respectively. The results showed that, when compared to the controls (FT+EtoH, TIR1+EtoH, TIR1+IAA), the replication of parasites was not affected obviously in FT+IAA group (Figure 4a). Nevertheless, it was clearly observed that the number of vacuoles with two or four parasites was apparently increased, and the vacuoles with eight parasites were significantly decreased in GGT−1+IAA group when compared with controls (GGT-1+EtoH, TIR1+EtoH, TIR1+IAA) (*p* < 0.001) (Figure 4a). The GGT-2+IAA group presented a similar trend as the GGT-1+IAA group, but it displayed more obvious effects on the replication of parasites than the GGT-1+IAA group (*p* < 0.001) (Figure 4b). 

### 2.5. Effects of FT, GGT-1 and GGT-2 on Gliding Ability of T. gondii 

To further explore the functions of FT, GGT-1 and GGT-2 on *T. gondii*, we examined the gliding ability of parasites when FT, GGT-1 and GGT-2 were degraded. The present study demonstrated that parasites could produce normal moving tracts in response to the degradation of FT, and GGT-1 and did not show obvious differences when compared with the controls (TIR1±IAA, FT-IAA, GGT-1-IAA) (Figure 5a). However, degradation of GGT-2 in parasites led to abnormal moving tracts when compared to the control groups (TIR1±IAA, GGT-2-IAA) under the same treatment conditions. Furthermore, the length of moving tracts in GGT-2+IAA group was significantly decreased when compared to the controls by statistical analysis (*p* < 0.001) (Figure 5b).

### 2.6. Effects of FT, GGT-1 and GGT-2 on Organelles of T. gondii

In the current study, the phenotypes of rhoptry, microneme, inner membrane complex (IMC), mitochondrion, centrin, apicoplast, microtubes and endoplasmic reticulum (ER) were examined by IFA after the degradation of FT, GGT-1 and GGT-2, respectively. The markers ROP5, MIC2, IMC1, Hsp60, Cen1, ACP, Tubulin and Bip symbolized rhoptry, microneme, IMC, mitochondrion, centrin, apicoplast, microtubes and ER, respectively. 

When FT was degraded, the localization of ROP5 became from regular sticks shape in apical area to disorderly spots in apical and middle of parasites (Figure 6a). However, the morphology of microneme, IMC, mitochondrion, centrin, apicoplast, microtubes and ER was not changed when compared to the control group (FT-IAA) in response to the degradation of FT (Appendix A). After the depletion of GGT-1, the phenotype of rhoptry was also altered, which was similar to that in FT (Figure 6b). Additionally, the mitochondrion produced discontinuous fractures and absence (Figure 6d). However, the other organelles (microneme, IMC, centrin, apicoplast, microtubes and ER) were not affected when GGT-1 was degraded (Appendix A). The degradation of GGT-2 also affected the phenotype of rhoptry, where ROP5 dispersed in the whole cytoplasm (Figure 6c). Furthermore, the microtubes and IMC displayed fractures and disorders after GGT-2 was degraded (Figure 6g,h). The absence of GGT-2 did not cause the obvious changes in centrin, apicoplast, mitochondrion, microneme and ER (Appendix A).

The distribution of GRA7 was also detected when the FT, GGT-1 and GGT-2 were degraded, respectively. The results illustrated that GRA7 mislocalized in the internal parasites after the degradation of GGT-2 and GGT-1, respectively, when compared to the controls (GGT-2-IAA, GGT-1-IAA), where GRA7 localized surrounding the parasites (Figure 6e,f). However, the depletion of FT did not lead to changes in the localization of GRA7 (Appendix A).

### 2.7. GGT-2 Is Essential for the Organization of Subpellicular Microtubules in Mature Parasites

The former results demonstrated that the morphology of microtubes was disorganized, identified by the intracellular IFA after the depletion of GGT-2 (Figure 6g). In order to observe intuitively whether the microtubular cytoskeleton would be affected by GGT-2, the parasites were extracted by deoxycholate and stained with anti-tubulin antibody. It could be clearly shown that the degradation of GGT-2 led to the apparent disorganization of subpellicular microtubules (Figure 7a). Furthermore, through the statistical morphology of subpellicular microtubules, the abnormal morphology of microtubes was dramatically increased after the depletion of GGT-2 when compared to the controls (GGT-2+EtOH, TIR1+EtOH, TIR1+IAA) (Figure 7b). However, the degradation of FT and GGT-1 did not cause obvious morphology defects in subpellicular microtubules when compared to the control groups (FT+EtOH, GGT-1+EtOH, TIR1+EtOH, TIR1+IAA) (Figure 7b).

## 3. Discussion

The determination of proteins’ localization could provide essential information to analyze their functions in parasites. For example, the kinase-extracellular-signal-regulated kinase 7 (ERK7) of *T. gondii* localized in the apical cap of parasites and was identified to be essential for the biogenesis of apical complex and function at the last step of conoid biogenesis [30]. The protein TgApiAT5-3, which is distributed in the plasma of parasite, was found to act as a transporter to exchange aromatic and large neutral amino acids, especially for the transportation of L-tyrosine [31]. In the current study, FT, GGT-1 and GGT-2 were recognized to distribute in the cytoplasm of parasites, which provided an important clue that these proteins might participate in multiple activities in parasites. 

With the aim of clarifying the effects of FT, GGT-1 and GGT-2 on *T. gondii* and considering the crisper values of GGT-1 (−4.25) and GGT-2 (−4.27), the auxin-inducible degron (AID) system was used in this study. The AID system has been used to investigate the functions of *T. gondii* proteins in several studies. For example, *T. gondii* conoid protein hub 1(TgCPH1) labelled with AID degron is essential for the conoid’ structural integrity and invasion and motility ability of parasites [32]. The Kelch13 protein tagged with AID was proved as a hub protein of the micropore structure and coordinated the uptake of host cytoplasmic metabolites and Golgi metabolites with other micropore proteins [33]. In this study, FT-AID, GGT-1-AID and GGT-2-AID could be effectively degraded after adding IAA for 24 h, which could be evidently observed in Figure 2. The successful construction of conditional strains (FT-AID, GGT-1-AID and GGT-2-AID) supplied a foundation for later investigations. 

The replication of parasites was examined in the current study. The results proved that the degradation of GGT-1 and GGT-2 largely inhibited the replication of parasites. Previous studies reported some mechanisms regarding the replication of *T. gondii*. For example, the inner membrane complex 32 (IMC32) of *T. gondii* depended on both a palmitoylation site of the N-terminal and some coiled-coil domains of C-terminal to function during replication and identified to be essential for parasite replication [34]. TgRab11a was phosphorylated by TgCDPK7 and then played key roles in parasite division [35]. The current study suggested that prenylation modification might participate in the process of parasite replication. However, the molecules modified by GGT-1 and GGT-2 functions in the replication process of parasites are still unknown. 

Gliding motility was essential for parasites to migrate through tissues, across biological barriers and invade host cells [36]. Kristen M. Skillman et al. clarified that the inherent instability of actin filaments of *T. gondii* was an essential adaptation for gliding motility [37]. The acto-myoA motor complex was proved to be important but not vital for the gliding motility of *T. gondii* [38]. Some proteins modified by palmitoylation modification play significant roles in the gliding of *T. gondii* [39]. *T. gondii* actin depolymerizing factor (TgADF) was verified to be mandatory for the process of rapid turnover of actin filaments and gliding motility in parasites [40]. The results of this study showed that GGT-2 was essential for the gliding motility of *T. gondii*, which first uncovered that prenylation modification was required for the process of gliding motility. However, the mechanism for the participation of GGT-2 needs to be further investigated.

The phenotype of multiple organelles of *T. gondii* was observed after the degradation of FT, GGT-1 and GGT-2, respectively. The results apparently presented that the morphology of rhoptry was largely affected when the three prenyl transferases were degraded. The former studies reported that some Rab proteins participated in the pathways of vacuolar sorting and biogenesis of rhoptry and microneme, such as Rab5 and Rab7 [41,42]. However, whether these rab proteins functioning in vacuolar sorting are regulated by prenyl transferases requires further investigation. The current results illustrated that deletion of GGT-1 affected the morphology of mitochondrion and degradation of GGT-2 caused the disorder of microtubes and IMC. Nevertheless, the proteins involved in these processes modulated by GGT-1/GGT-2 are largely unknown and should be examined in future studies. Notably, the localization of GRA7 was mislocalized when GGT-1 and GGT-2 were degraded. The previous studies reported that GRA7, as one parasitophorous vacuole membrane-associated dense granule protein, played significant roles in the development of chronic-stage cysts in vivo [43] and could regulate the maturation of cyst wall of *T. gondii* [44]. The current study suggests that GGT-1 and GGT-2 may affect the functions of GRA7 by altering its localization. The molecular mechanism involved in the process needs to be further investigated.

In a word, FT, GGT-1 and GGT-2, as prenyl transferases, could modify the proteins needed to be prenylated. The prenylated proteins might participate in the vacuolar sorting to multiple organelles. The current study indicated that FT, GGT-1 and GGT-2 might be involved in the process of vacuolar sorting to rhoptry. Vacuolar sorting into inner membrane complex and mitochondrion might be related to GGT-2 and GGT-1, respectively (Figure 8).

## 4. Materials and Methods

### 4.1. Plasmids, Parasites and Cells

The plasmids pLinker-6HA-HXGPRT, pSAG1:CAS9-U6:sgUPRT and pLinker-AID-6Ty-DHFR were stored our lab. The RHΔ*ku80*Δ*hxgprt* line was obtained from Prof. Vern Carruthers. Human foreskin fibroblast (HFF) cells were purchased from American Type Culture Collection (ATCC). The TIR1 parasite line and murine ty hybridoma cells (generation of the Ty monoclonal antibody) were kindly donated by L. David Sibley from Washington University in St. Louis, USA.

### 4.2. Conservation and Phylogenetic Analysis

The prediction sites within prenyltrans PFam domains for the three genes were queried according to the OrthoMCL database (https://orthomcl.org/orthomcl/app/) (accessed on 2 August 2022). In order to explore the phylogenetic relationship between FT, GGT-1 and GGT-2 of *T. gondii* and organisms from the Alveolate, the amino acid sequences were used as queries for searching the protein homologues with the iterative version of the profile hidden Markov models (HMMs) search engine (jackhammer) (E-value cutoff for the best hit: 1 × 10^−7^). The best hit of representative species was selected from the remaining candidate taxa for further phylogenetic analysis. Phylogenetic analysis was performed based on the aligned amino acid sequences of the above species. Sequences were firstly aligned using Mafft (v7.490, Kyoto, Japan) software with the L-INS-I algorithm. The spurious sequences or poorly aligned regions were automatedly removed using trimAL software (https://vicfero.github.io/trimal/) (v1.2, Barcelona, Spain) (accessed on 10 August 2022) according to the “automated1” parameter. Phylogenetic tree was constructed by FastTree (v2.1, Berkeley, CA, USA) using the Le-Gascuel (LG) model. Finally, the obtained maximum likelihood tree was displayed via Chiplot online portal (https://www.chiplot.online/) (accessed on 20 August 2022). The heatmap of conservation analysis was further visualized by the Complex Heatmap package in R (v4.2.0, Auckland, New Zealand). 

### 4.3. Construction of FT-6HA, GGT-1-6HA and GGT-2-6HA Parasites

The HA tag was added to the C-terminal of FT, GGT-1 and GGT-2 using genome-edit technology based on the CRISPR−Cas9 system. The completion of constructing FT-6HA, GGT-1-6HA and GGT-2-6HA strains could totally include three steps. In brief, the designation and construction of C-terminal sgRNA of FT, GGT-1 and GGT-2, the amplification of homologous fragments, parasite transfection and screening of positive clones. The principles and methods could refer to the former descriptions [45]. Firstly, the web server E-CRISP (http://www.e-crisp.org/E−CRISP/designcrispr.html) (accessed on 4 June 2022) was used to design specific gRNA sequences that targeted the FT, GGT-1 and GGT-2. Here, the gRNA sequence lied in the upstream of PAM sequence (NGG) in the 3′ UTR of the genome and was a total of 20 bp in length. The detailed information could be shown in Appendix A. After selecting the proper gRNA, the sgRNA targeting UPRT in the original CRISPR/Cas9 plasmid (pSAG1::Cas9-U6::sgUPRT) is replaced with the selected gRNA by PCR homologous recombination. The homologous plasmids were then transferred into *E. coli* (DH5α). Subsequently, the clones were selected from the LB medium plate containing 100 μg/mL ampicillin and then identified by DNA sequencing using the primer M13R (5′-CAGGAAACAGCTATGAC).

The amplification of homologous fragments should include 5′ homology arms (42 bp) upstream of the stop codon, an epitope tag sequence followed by one drug selection cassette (6HA-HXGPRT) and 3′ homology flanks (42 bp) downstream of the stop codon. The primers designed for amplifying homologous fragments can be found in Appendix A. 

The CRISPR plasmid containing specific sgRNA and the corresponding homologous fragments were electroporated into the background strain RHΔ*ku80*Δ*hxgprt*. After that, the parasites were selected in D5−containing drugs (mycophenolic Acid, xanthine) and finally examined by immunofluorescence assay using anti-HA antibody.

### 4.4. Construction of Conditional Knock-Down Strains of FT, GGT-1 and GGT-2 

In this study, we utilized the auxin-inducible degron (AID) system to investigate the effects of FT, GGT-1 and GGT-2 on *T. gondii*. The protocol of AID system could refer to the reported methods [46]. Simply stated, the degron AID was added to the C-terminal of FT, GGT-1 and GGT-2. The ubiquitin ligase complex SCFTIR1 was activated once it was treated with 3-indolacetic acid (IAA). Afterward, SCFTIR1 could exclusively target AID-tagged FT, GGT-1 and GGT-2 and then the process of ubiquitin-dependent proteasomal degradation was started [46]. The construction of FT-AID, GGT-1-AID and GGT-2-AID strains was similar to that of FT-6HA, GGT-1-6HA and GGT-2-6HA strains. The differences were mainly in three aspects: the background strain (RHΔ*ku80*Δ*hxgprt*; TUB1:TIR1-3FLAG, SAG1:CAT), the amplification of homologous fragment (AID-6TY-DHFR) and drug selection (pyrimethamine). The primers related to the whole process could refer to Appendix A.

### 4.5. Indirect Immunofluorescence Assay

As for *T. gondii*, indirect immunofluorescence assay (IFA) was an important tool for examining the proteins’ localization, degradation, the parasites’ replication and gliding abilities. The protocols of IFA were described briefly as follows [32]:

The influent HFF cells were infected with *T. gondii* in T-25 flasks to reach around 70–85% host cell lysis after 2 days’ infection. The monolayers were scraped and then passed by one 22 G blunt-end needle about 4 times to disrupt HFF cells and let all parasites release. Infecting the influent HFF cells with the released parasites (1 × 10^5^) in 24-well plates with cell crawling piece (Bochuangsheng, Beijing, China). After 24 h of incubation in one 37 °C 5% CO_2_ incubator, the culture medium in plates was discarded. Subsequently, the plates were washed with warm PBS twice and fixed with 4% polyformaldehyde (PFA) (Solarbio, Beijing, China) for 15 min. PFA was removed and the plates were supplied with 0.25% TritonX-100 (diluted with 2.5% bovine serum albumin) (Solarbio, Beijing, China). After 15 min permeabilizing, TritonX-100 was abandoned and plates were incubated with 2.5% BSA (Solarbio, Beijing, China) for 40 min. Next, the cell crawling pieces were incubated with the related primary antibodies (diluted with 2.5% bovine serum albumin) for 40 min, washed by 0.05% Tween-20 diluted with 2.5% bovine serum albumin (washing buffer) five times and each time for 5 min. Afterward, secondary antibody incubation for 30 min and the same washing procedure were executed followed by DAPI (Solarbio, Beijing, China) staining and mounting medium fixation. Finally, the cell crawling pieces were observed under a laser-scanning confocal microscope.

### 4.6. Parasite Replication Assay

*T. gondii* replicates within parasitophorous vacuole in host cells in the form of binary fission [47]. There are several parasite forms within parasitophorous vacuoles, such as 1, 2, 4, 8, 16 and so on. The protocol of replication assay in this study utilized the basic IFA operations with some modifications. The details were as follows: 

Firstly, the preparation of parasites was the same as that in IFA. The prepared parasites about 1 × 10^5^ (15–20 μL parasite suspension) were added to the influent HFF cells in 24-well plates with cell crawling pieces. During 3 h’ incubation, the culture medium (DMEM with 5% fetal calf serum, D5) was discarded and the cell crawling pieces were supplemented with fresh D5 containing 500 μM IAA (the experimental group). Contrary to the experimental group, cell crawling pieces without IAA (Sigma, St. Louis, MO, USA) were regarded as the negative group. Afterward, the plates were placed in an incubator at 37 °C 5% CO_2_ for 21 h. After that, the cell crawling pieces were operated by 4% PFA fixation, 0.25% TritonX-100 permeabilization, incubation of primary antibody (anti-mouse HA) and secondary antibody (Alexa FluorTM 488 goat anti-mouse), DAPI staining and mounting medium fixation, respectively. Lastly, the parasites were observed in a laser-scanning confocal microscope and at least 200 parasitophorous vacuoles of each group were analyzed. 

### 4.7. Parasite Gliding Assay

The 24-well plates with cell crawling pieces were supplemented with 500 μL blocking buffer (5% BSA) and then placed in a 4 °C refrigerator for one night. Subsequently, the blocking buffer was removed and the plates were washed with warm PBS followed by adding warm EC buffer (5.6 mM D-glucose, 25 mM HEPES, 1 mM MgCl_2_, 1.8 mM CaCl₂, 142 mM NaCl, 5 mM KCl, PH: 7.4). The plates were put in 37 °C 5% CO_2_ for preparation. Preparing the parasites not egress or little egress from HFF cells and the monolayers were scraped and then passed by one 22 G blunt-end needle about 4 times to disrupt HFF cells and let all parasites release. Parasites were then filtered (4 μm filter), centrifuged, washed and resuspended with EC buffer, respectively. The EC buffer (500 μL) containing parasites (5 × 10^6^) was added to the prepared plates in the former step. Next, the plates were transferred in a constant temperature bath (37 °C) for 15 min. The parasites’ suspension in plates was discarded, followed by washing with warm PBS, fixing with 4% PFA for 15 min and incubating with 5% BSA for 40 min, respectively. After that, the cell crawling pieces were incubated with a primary antibody (mouse anti-Toxoplamsa SAG1 monoclonal antibody) and followed by a secondary antibody (Alexa FluorTM 488 goat anti-mouse) (ThermoFisher Scientific, Waltham, MA, USA). After mounting medium fixation, the cell crawling pieces were observed in a laser scanning confocal microscope (Nikon, Nikon Instruments, Shanghai, China) [48]. Different moving tracts were calculated, for example, gliding, revolving and so on. The software NIS−Elements AR (Nikon, Nikon Instruments, Shanghai, China) was used to calculate the length of moving tracts and 20–40 parasites’ moving tracts were analyzed. 

### 4.8. Deoxycholate Extraction

The 24-well plates with cell crawling pieces coated with poly-L-lysine were first prepared. After HFF cells were infected with the parasites treated with IAA or EtOH for 24 h, the monolayers were disrupted and fresh parasites were deposited on the coverslips. Afterward, the coverslips with parasites were treated with 10 mM deoxycholate (Sigma, St. Louis, MO, USA) for 20 min at room temperature, followed by fixing with cold methanol for 8 min. Next, IFA was performed as previously described [49]. In the assay, the primary antibody was anti-rabbit tubulin and the secondary antibody was Alexa FluorTM 488 goat anti-rabbit. 

### 4.9. Statistical Analysis

Statistical analyses were implemented by GraphPad 8.0 software (GraphPad Prism, San Diego, CA, USA). The data acquired from the above experiments were presented as means ± SD. The analysis of variance (ANOVA) was used to illustrate the differences among groups with **** p* < 0.001.

## 5. Conclusions

In summary, the FT, GGT-1 and GGT-2 of *T. gondii* played significant roles in the replication and gliding motility of parasites. Furthermore, the results showed that proteins modified by FT, GGT-1 or GGT-2 might participate in vacuolar sorting to different organelles. The study first clarified the effects of FT, GGT-1 and GGT-2 on *T. gondii* in detail, which benefited the understanding of prenyltransfereases and the discovery of new drug targets.

## Figures and Tables

**Figure 1 ijms-24-07172-f001:**
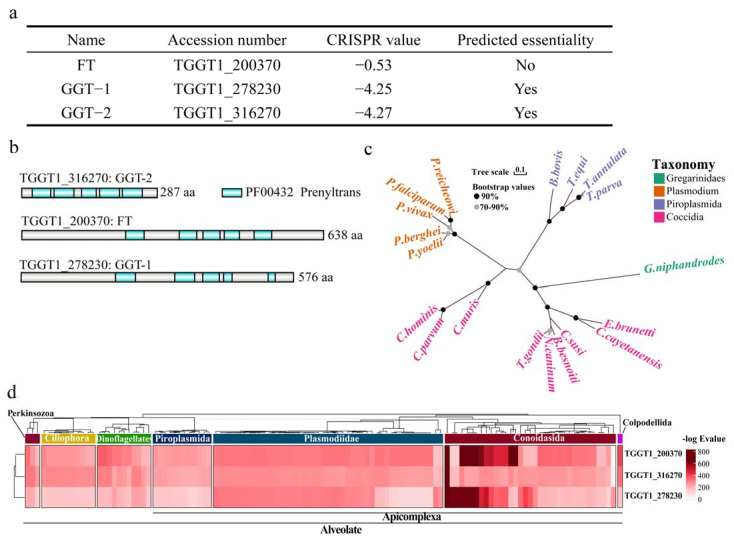
Conservation and phylogenetic analysis for FT, GGT-1 and GGT-2. (**a**) The information about FT, GGT-1 and GGT-2 could be obtained from the website (https://toxodb.org/toxo/app) (accessed on 2 June 2022). (**b**) Schematic diagram of domains in prenyl transferases. (**c**) Maximum likelihood tree presented the phylogenetic relationships of prenyl transferase homologs in diverse apicomplexans, and the bootstrap values of the branch greater than 70% were displayed. The scale bar indicated the genetic distance. (**d**) Heatmap indicating conservation of associated proteins among Alveolata. The tree dendrogram without meaningful branch lengths across these species and identified proteins were clustered according to Euclidean distance using the UPGMA method.

**Figure 2 ijms-24-07172-f002:**
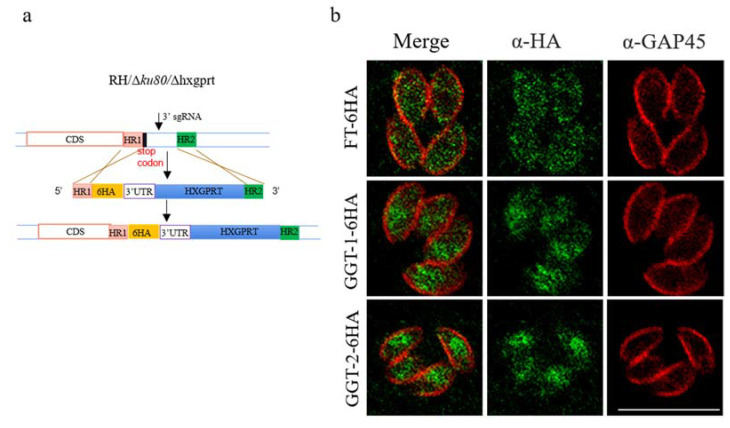
FT, GGT-1 and GGT-2 are localized in the cytoplasm of parasites. (**a**) The model of target genes (FT, GGT-1 and GGT-2) tagged with HA at the C-terminal under the background strain RHΔ*ku80*Δ*hxgprt* using genome-editing technology. (**b**) IFA was used to identify the localization of FT, GGT-1 and GGT-2. GAP45 was used to label the outer plasma of the parasite (red color) and the target proteins were identified by ant-HA antibody (green color). Scare bar: 10 μM.

**Figure 3 ijms-24-07172-f003:**
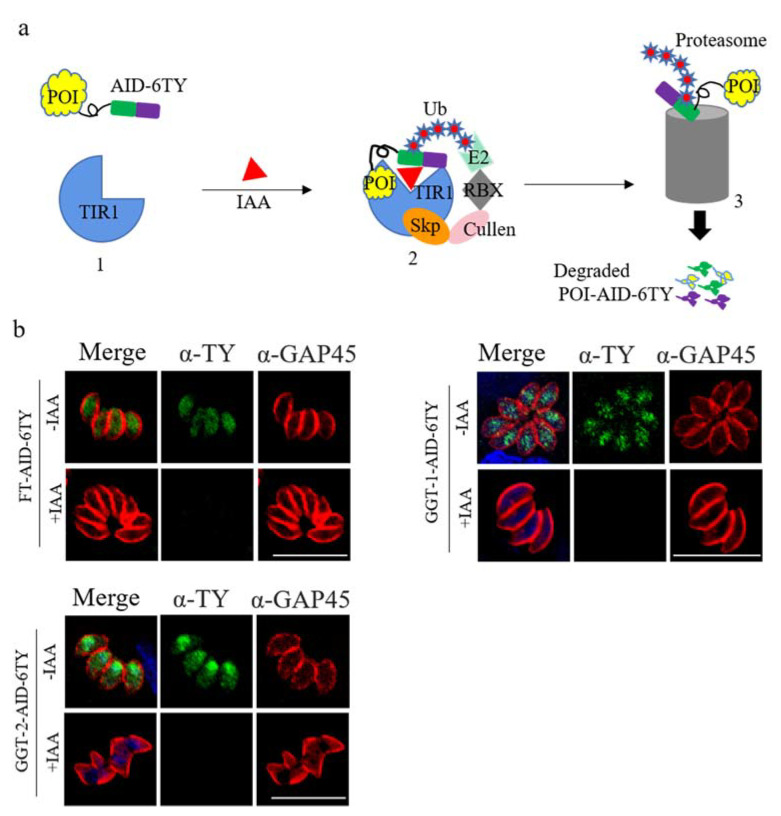
FT, GGT-1 and GGT-2 were degraded using the TIR1-AID system. (**a**) Model of target proteins degraded by TIR1-AID system. In this system, the receptor TIR1 is in inactive state in the absence of IAA, permitting the fusion protein POI-AID-6TY to function in a normal way (1). After adding IAA, TIR1 can be activated, which prompts the formation of Skp-Cullen-F Box (2). The Box can identify and polyubiquitinate AID, leading to the proteasomal degradation for POI-AID-6TY (3). POI: protein of interest. (**b**) FT-AID-6TY, GGT-1-AID-6TY and GGT-2-AID-6TY could be effectively degraded after adding IAA for 24 h. The degradation of FT, GGT-1 and GGT-2 was detected by IFA. Parasites were stained with anti-mouse ty monoclonal antibody (green) and anti-rabbit GRA45 (red). Nuclei were stained with DAPI (blue). Scare bars: 10 μM.

**Figure 4 ijms-24-07172-f004:**
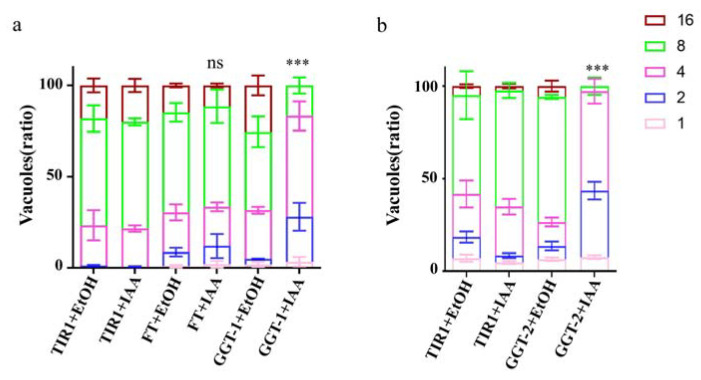
GGT-1 and GGT-2 regulated the replication of parasites. (**a**) The replication analysis of FT and GGT-1. (**b**) The replication assay of GGT-2. IFA was carried out after the parasites grew in HFF with IAA for 24 h. GAP45 was utilized to stain the outline and at least 200 parasitophorous vacuoles were calculated. Data were representative of three independent experiments and the values presented the means ± SEM (**** p* < 0.001, ns: not significant).

**Figure 5 ijms-24-07172-f005:**
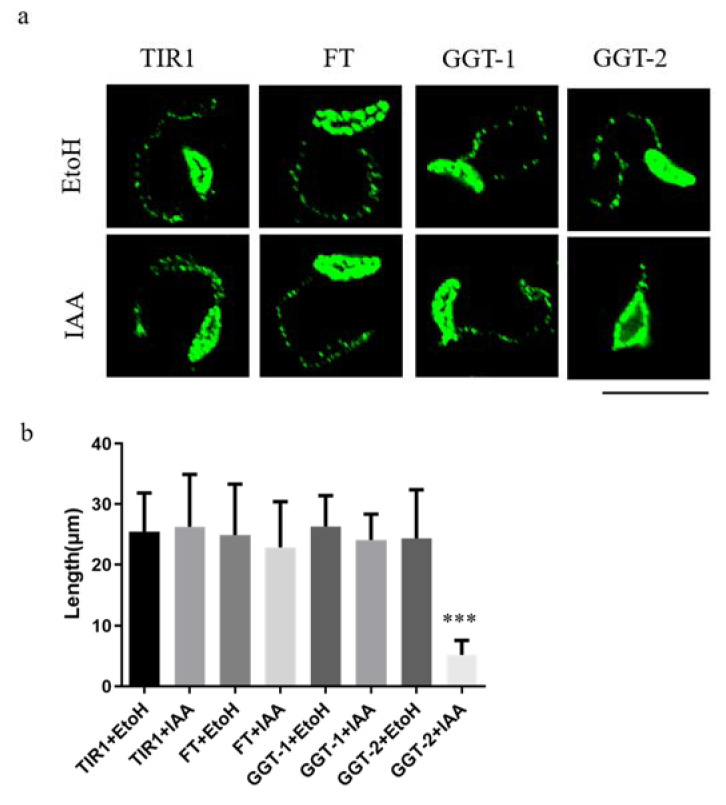
Degradation of GGT-2 inhibited the gliding motility of parasites. (**a**) The observation of moving tracts after TIR1/FT/GGT-1/GGT-2 was treated with IAA or EtoH for 24 h. IFA was performed when parasites were treated with 10 mM deoxycholate for 20 min and a mouse anti-Toxoplamsa SAG1 monoclonal antibody was used as the primary antibody (green). (**b**) At least 20 moving tracts lengths of parasites were quantified in this assay. The data were presented as the means ± SEM with triplicate experiments (**** p* < 0.001). Scare bar: 10 μM.

**Figure 6 ijms-24-07172-f006:**
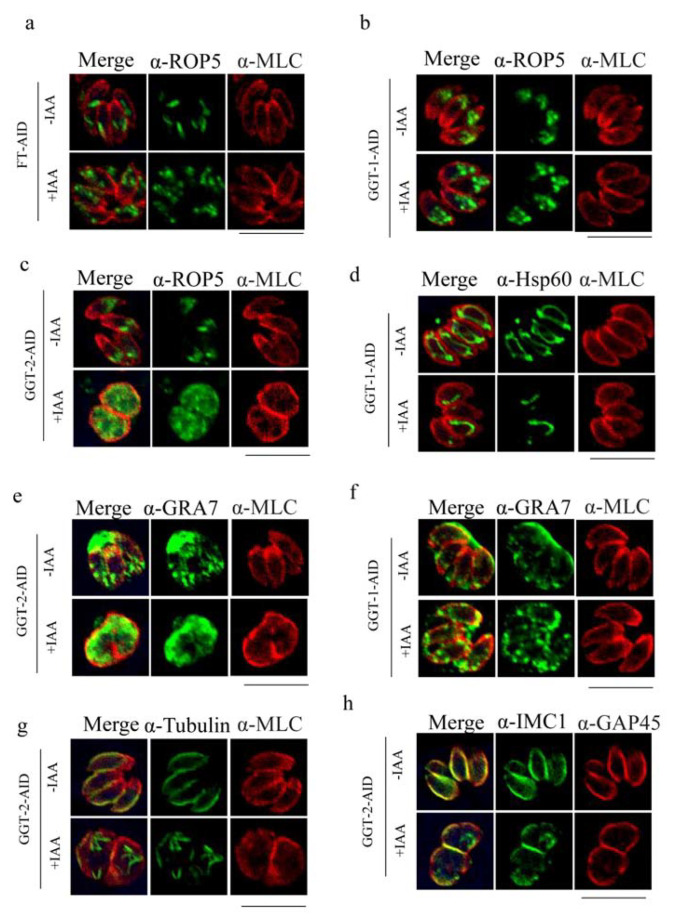
Degradation of FT, GGT-1 and GGT-2 affects the morphology of multiple organelles of parasites. (**a**) Effects of absence of FT on rhoptry. Effects of degradation of GGT-1 on rhoptry (**b**), mitochondrion (**d**) and GRA7 (**f**). Observation of morphology changes of rhoptry (**c**), GRA7 (**e**), microtubes (**g**) and inner membrane complex (**h**) after GGT-2 was degraded. IFA was performed to observe the morphology of organelles. MLC or GAP45 was used to label the outer plasma of parasite (red), target proteins were stained with green color, and the yellow color symbolized the merged localization between the target protein and the outer plasma of parasite in this assay. ROP5, Hsp60, Tubulin and IMC1 were symbolized as the markers of rhoptry, mitochondrion, microtubes and inner membrane complex, respectively. Scare bars: 10 μM.

**Figure 7 ijms-24-07172-f007:**
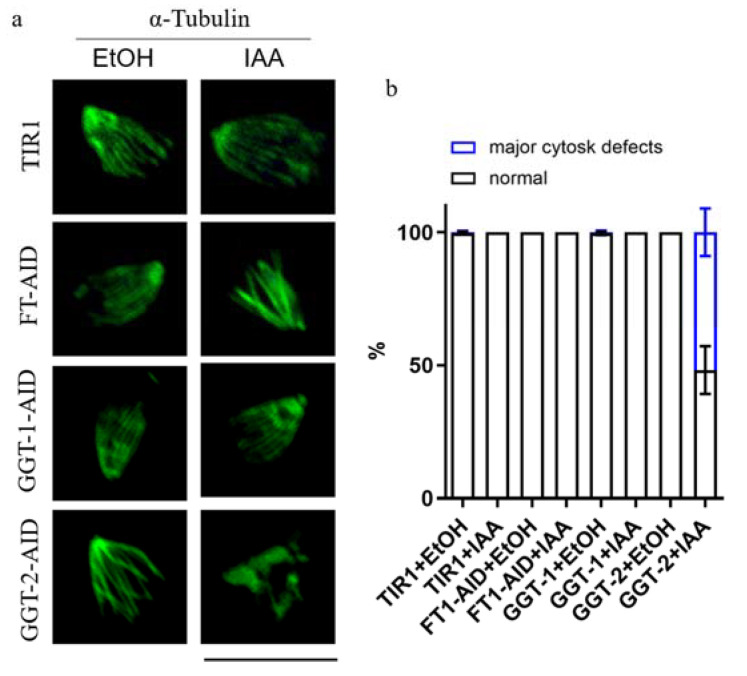
GGT-2 affects the subpellicular microtubules of parasites. (**a**) IFA was used to observe the effects of FT, GGT-1 and GGT-2 on subpellicular microtubules. The anti-rabbit tubulin and Alexa FluorTM 488 goat anti-rabbit were used as the primary and secondary antibodies, respectively. (**b**) The normal and abnormal morphology of microtubules in parasites was quantified, respectively, and at least 50 parasites’ subpellicular microtubules were analyzed. Scare bar: 10 μM.

**Figure 8 ijms-24-07172-f008:**
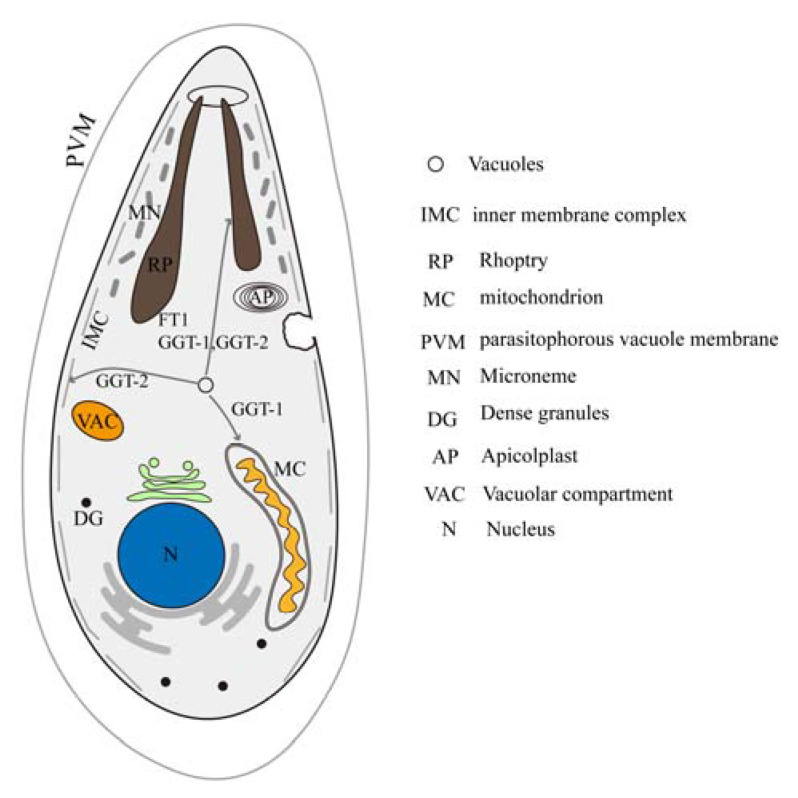
Model of the effects of FT, GGT-1 and GGT-2 on organelles of the parasite. IMC: inner membrane complex; RP: rhoptry; MC: mitochondrion; PVM: parasitophorous vacuole membrane; MN: microneme; DG: dense granules; AP: apicoplast; VAC: vacuolar compartment; N: nucleus.

## Data Availability

The datasets supporting the conclusions of this article are included within the article and its additional files.

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
