# Peer review of "Prenyl Transferases Regulate Secretory Protein Sorting and Parasite Morphology in Toxoplasma gondii"

_ijms, 2023, doi:10.3390/ijms24087172_

Round 1
Reviewer 1 Report
Authors applied endogenous tagging and AID knock-down techniques to investigate three putative prenyltransferases in Toxoplasma gondii, the experiments are well designed, results are clear and conclusive. However, still some important points need address.
1, Line 17 & 22 and section 2.6, "phenotype" or just morphology, or mis-localization of a set of proteins.
Question: Is these proteins need prenylation modification? Please discuss it.
2, Fig4a&b, the controls (TIR1) seems the same, if it is true, suggestion to merge these two panels.
3, Section 2.8, suggest to move to discussion.
Minor points:
Line 16 & 17, delete three "the", as knockdown and auxin-induciable appear the first time, same as the "phenotype".
Line 27&28, change to "provided the theoretical basis for future drug design"
Line 34, delete "(T. gondii)"; change "the" into "a";
Line 35&36, delete "as the world’s most suc-35 cessful parasite,";
change to "it can infect almost all nucleated cells of warm-blooded animals"
Line 37, change to "Although the parasite generally cause mild symptoms in immunocompetent people"; delete "new";
Line 46&47, please explain why "these chemotherapeutic choices for toxoplasmosis are very limited"; there is no "development" of "drug targets", same as line 71, suggest to change to "identification".
Line 50&51, change to "need enzymes of three types:"
Fig1b,c&d, words are too small to read.
change to "(a) The information about FT, GGT-1, and GGT-2 from the website (https://toxodb.org/toxo/app)"
change to "(b) Schematic diagram of domains in prenyltransferases"
Fig4a&b, the controls (TIR1) seems the same, if it is true, suggestion to merge these two panels.
Line 162&410, "anti-mouse SAG1 monoclonal antibody" I think it should be "mouse anti-Toxoplamsa SAG1 monoclonal antibody"
Reviewer 2 Report
In this study, the authors studied the important biological functions of three prenyltransferases of T. gondii (TgFT, TgGGT-1, TgGGT-2). This study is interesting. Some minor issues should be clarified before its acceptable for publication.
1. At lines 32-34, “The Apicomplexa includes many intracellular parasitic protozoa, which can lead to some important diseases, including malaria, cryptosporidiosis, toxoplasmosis, coccidiosis, theileriosis, babesiosis and so on.” should be deleted.
2. For the figure 6d, GGT-1-AID, what's the difference between the two IAA treatment?
3. At line 300, what is the use of mouse ty hybridoma cells? Specific information should be given.
Reviewer 3 Report
Review of Manuscript ijms-2306716: Prenyltransferases regulate multiple organelles morphology, intracellular replication, and gliding motility in Toxoplasma gondii by Wang, He, Aleem and Long.
Summary: In this manuscript the authors investigated protein prenyltransferases in T. gondii (Tg) and examined the effects of three prenyltransferases (TgFT, TgGGT-1, TgGGT-2) on various organelles of T. gondii and on parasite replication and gliding ability, using a combination of immunofluorescence assays (IFA) and conditional knockout parasites generated via the auxin-inducible degron fusion based upon CRISPR/Case9. Major results included effects on the morphology of rhoptry, IMC and mitochondrial organelles and the parasitophorous vacuole membrane (PVM), a reduction in parasite replication and gliding motility when Tg prenyltransferases were deleted. Specifically, TgFT affected on the morphology of rhoptry, TgGGT-1 affected the morphology of rhoptry, mitochondrion, and PVM, while degradation of TgGGT-2 destroyed the PVM, IMC and rhoptries. TgGGT-1 and TgGGT-2 affected parasite replication and TgGGT-2 weakened the gliding ability of parasites. There is little prior knowledge of Tg prenyltransferases and this study provides new information on the functional importance of the prenyltransferases in Tg that may contribute to drug targets in T. gondii. The manuscript consists of 8 Figures.
Overview: The manuscript contributes knowledge about protein prenyltransferases in T. gondii, which have been shown to have promise as a drug target in malaria, but which have not been studied in T. gondii. In this manuscript three protein prenyltransferases were found to have effects on various organelles and to affect parasite motility and replication. The strengths of the manuscript are contribution to the literature on prenyltransferases in Toxoplasma and the experimental approach of using auxin induced conditional knockdowns to address the functions of these enzymes. The limitation of this manuscript is that while effects of the three prenyltransferases were found, the protein targets of these enzymes were not identified or in any way defined. In addition, there are issues with writing including some over-interpretation or lack of some supporting data, inaccurate or incomplete statements of the Toxoplasma literature, and some incorrect or imprecise uses of English, as specified in the comments below.
1. Title: The use of the term ‘regulate’ in the title seems too strong a word for the data presented with the manuscript.
2. Abstract:
Line 14 – In the sentence beginning, ‘However, seldom’, the word seldom would be better replaced with ‘few’ as the authors refer to the number of studies
3. Introduction
Line 34- the phrase ‘so on’ ; a better phrase to use would be ‘among others’
Lines 34 -36 – The sentence beginning with the ‘Toxoplasma gondii’ mentions the parasite has ‘no strict host cell or cell type specificity’ which is largely true for the tachyzoite stage but only partially true for the bradyzoite stage in which the parasite is primarily in neurons in the brain and muscle cells in muscle tissue. Additionally, the phrase later in the sentence ‘it can infect the nucleated cells’ is a more accurate description of host cell specificity, and the inclusion of both phrases in the same sentence is redundant. Finally in this sentence the use of the phrase ‘the world’s most successful parasite’ is imprecise language which is inappropriate for a journal article. More specific information such as ‘a third of the world’s population’ or something to that effect would be more appropriate. Thus, this sentence would be best rewritten with all the above issues addressed.
Line 46-47 – In the sentence beginning with the word ‘However…’ it is not clear
what, the word ‘limited’ refers. Do the authors mean clinically approved drugs? And if so, what is the problem with current drugs. Or are the authors referring to something else?. As the potential application for results of this study are potential new drug targets, clarification of this point is Important for the paper.
4. Results
Line 98-105 – On the localization of FT, GGT-1, GGT-2, the authors state all three are found in the cytoplasm. While the IFA data is convincing, is there any supporting data such as co-localization with a known cytoplasmic marker or cell fractionation, etc., to support this conclusion?
Line 113 – Sentence beginning with the word ‘The AID system’ the acronym AID should be defined before use as this is the first place in the manuscript using this term.
Line 120-127 – Regarding Fig. 3, the acronym, POI should be defined in the Fig. legend.
Line 222-6 – The authors use the word ‘vacuoles’ to refer to sorting into multiple organelles such as the rhoptry and IMC. In T. gondii the term ‘vacuole’ usually refers to the parasitophorous vacuole (PV) which is not the intent here and is thus confusing. The phrase ‘vacuolar sorting’ if this is what the authors are referring to, would clarify the meaning.
5. Discussion
Line 246 – the phrase ‘widely used’ in the sentence beginning with ‘For example’ referring to the use of AID system, yet only 1 reference is cited. Either more references are needed to support this statement or a softer word such as ‘several’ could be used.
6. Conclusions
Line 434 -use of the phrase ‘vacuoles sorting’ is confusing, similar to the comment referencing line 246.
